# Cinnamal Sensing and Luminescence Color Tuning in a Series of Rare-Earth Metal−Organic Frameworks with Trans-1,4-cyclohexanedicarboxylate

**DOI:** 10.3390/molecules26175145

**Published:** 2021-08-25

**Authors:** Pavel A. Demakov, Alena A. Vasileva, Sergey S. Volynkin, Alexey A. Ryadun, Denis G. Samsonenko, Vladimir P. Fedin, Danil N. Dybtsev

**Affiliations:** 1Nikolaev Institute of Inorganic Chemistry, Siberian Branch of the Russian Academy of Sciences, 630090 Novosibirsk, Russia; a.vasileva2@g.nsu.ru (A.A.V.); volynkin@niic.nsc.ru (S.S.V.); ryadunalexey@mail.ru (A.A.R.); denis@niic.nsc.ru (D.G.S.); cluster@niic.nsc.ru (V.P.F.); dan@niic.nsc.ru (D.N.D.); 2Department of Natural Sciences, Novosibirsk State University, 2 Pirogova St., 630090 Novosibirsk, Russia

**Keywords:** metal–organic frameworks, coordination polymers, X-ray diffraction analysis, rare-earth elements, luminescence, light-emitting devices, sensing, luminescent detection

## Abstract

Three isostructural metal–organic frameworks ([Ln_2_(phen)_2_(NO_3_)_2_(chdc)_2_]·2DMF (Ln^3+^ = Y^3+^ for **1**, Eu^3+^ for **2** or Tb^3+^ for **3**; phen = 1,10-phenanthroline; H_2_chdc = *trans*-1,4-cyclohexanedicarboxylic acid) were synthesized and characterized. The compounds are based on a binuclear block {M_2_(phen)_2_(NO_3_)_2_(OOCR)_4_} assembled into a two-dime nsional square-grid network containing tetragonal channels with 26% total solvent-accessible volume. Yttrium (**1**)-, europium (**2**)- and terbium (**3**)-based structures emit in the blue, red and green regions, respectively, representing the basic colors of the standard RGB matrix. A doping of Eu^3+^ and/or Tb^3+^ centers into the Y^3+^-based phase led to mixed-metal compositions with tunable emission color and high quantum yields (QY) up to 84%. The bright luminescence of a suspension of microcrystalline **3** in DMF (QY = 78%) is effectively quenched by diluted cinnamaldehyde (cinnamal) solutions at millimolar concentrations, suggesting a convenient and analytically viable sensing method for this important chemical.

## 1. Introduction

Metal–organic frameworks (MOFs) are an important class of coordination compounds extensively investigated during the last two decades. The porosity of MOFs could be systematically tuned in a wide range by variation of the topology and the length of the organic linker [1,2,3,4,5,6,7,8], while some of the physical properties of the framework mainly depend on the nature of metal centers [9,10,11,12,13]. Particularly, lanthanide coordination compounds are widely studied in magnetic and luminescent applications due to the unique electron configuration of the corresponding Ln(III) cations [14,15,16,17,18]. Luminescent porous MOFs containing lanthanide ions attract special interest as sensors for various ions or small molecules by tailoring both framework affinity towards certain substrates and analytical signal output. Examples of nitroaromatic compounds, activated arene derivatives and amide and carbonyl compound detection are most often reported, apparently due to intensive energy transfer processes possible between such guest molecules and the host framework [19,20,21,22,23,24,25,26].

Cinnamaldehyde (*trans*-3-phenylprop-2-enal, cinnamal) is an important component of cinnamon bark essential oil and frequently used in food flavoring and cosmetics. Cinnamaldehyde is also investigated as an insecticide, antifungal and antibacterial agent [27,28,29,30,31,32]. Despite generally low toxicity for human and occurrence in various common consumption goods, cinnamal is known to evoke allergic reactions [33]; therefore, its efficient detection is relevant for healthcare and other practical issues. Although there are a number of developed analytical methods for its quantitative determination [34,35,36,37,38,39,40], only one method of cinnamal fluorescent detection has been reported using nitrogen and sulfur codoped carbon dots [41]. To the best of our knowledge, no MOF-assisted techniques have been reported for cinnamal detection yet. 

In this work, a series of new isostructural rare-earth-based MOFs with the formulae [Ln_2_(phen)_2_(NO_3_)_2_(chdc)_2_]·2DMF (Ln^3+^ = Y^3+^, Eu^3+^, Tb^3+^, or their mixture) were synthesized and characterized. These compounds feature bright luminescence emission with tunable color and high quantum yields up to 32% for single-metal powder films and up to 84% for mixed-metal compositions. Luminescent turn-off response on cinnamal at low concentrations was also observed for microcrystalline dispersions of **3** in DMF solution, representing a first example of a MOF-based sensor for this important chemical. 

## 2. Results and Discussion 

### 2.1. Synthesis and Crystal Structure

A series of compounds [Ln_2_(phen)_2_(NO_3_)_2_(chdc)_2_]·2DMF (Ln^3+^ = Y^3+^ for **1**, Eu^3+^ for **2** and Tb^3+^ for **3**) was obtained at high yields by heating the mixture of metal nitrates, phenanthroline (phen) and *trans*-1,4-cyclohexanedicarboxylic acid (H_2_chdc) in *N*,*N*-dimethylformamide (DMF) solution at 110 °C for 48 h. The phase purity of **1**–**3** was established by powder X-ray diffraction (PXRD, Appendix A). Their chemical nature and composition were confirmed by CHN analysis, IR spectroscopy (Appendix A) and thermogravimetric analysis (TGA, Appendix A). IR spectra of **1**–**3** contain the characteristic bands of Csp^2^–H valence vibrations (3064–3086 cm^−1^), Csp^3^–H valence vibrations (2857–2937 cm^−1^), strong DMF CO stretchings (1678–1681 cm^−1^), coordinated carboxylate antisymmetric (1588–1598 cm^−1^) and symmetric 1425–1429 cm^−1^) stretchings and strong nitrate N–O stretchings (1291–1294 cm^−1^). All three compounds are isostructural; therefore, only the structure of the yttrium-based **1** will be described here in detail. **1** crystallizes in monoclinic symmetry with the *P*2_1_/*n* space group. The asymmetric unit contains one Y atom, one phen ligand, one coordinated NO_3_^−^ anion, one chdc^2−^ ligand and one DMF molecule. The metal center adopts a typical capped square antiprismatic environment (coordination number = 9). The Y–O(COO) distances are in the range of 2.2911(14)–2.5479(15) Å, the Y–O(NO_3_) distances are 2.4110(17) Å and 2.5101(17) Å and the Y–N(phen) distances are 2.5199(17) Å and 2.5683(19) Å. Two symmetrically equivalent Y^3+^ cations form a binuclear carboxylate block with the formula {Y_2_(phen)_2_(ONO_2_-κ^2^)_2_(μ-RCOO-κ^1^,κ^1^)_2_(μ-RCOO-κ^1^,κ^2^)_2_} (Figure 1a), interconnected by cyclohexanedicarboxylate bridging ligands into a 2D coordination network with a square-layered topology (Figure 1b). We should note here that blocks with either 2,2′-bipyridyl (bpy) or 1,10-phenantroline ligands are well known in the chemistry of lanthanide(III) complexes (69 unique hits in CSDB ver. 5.42 (November 2020) [42]) with monocarboxylate ligands, such as acetate, benzoate and their derivatives [43,44,45,46,47], but no examples for Y^3+^ have been reported so far. Additionally, **1**, as well as all other isostructural compounds obtained in this work, represents the first example of extended coordination polymer structures based on such a {M_2_L_2_(NO_3_)_2_(OOCR)_4_} building unit (where M = any metal cation, L = any chelate N-donor ligand). 

Recently, we reported a series of 3D coordination polymers [Ln_2_L_2_(chdc)_3_] (where Ln^3+^ = Y^3+^, Eu^3+^, Tb^3+^; L = bpy or phen) having dinuclear carboxylate building units {Ln_2_L_2_(OOCR)_6_} capped with two chelate N-donor ligands, which structurally resemble the building unit in **1** [48]. Both series [Ln_2_(phen)_2_(chdc)_3_] and [Ln_2_(phen)_2_(NO_3_)_2_(chdc)_2_] were obtained from similar reaction systems, starting from metal chlorides and metal nitrates, respectively. The poorly coordinating Cl^−^ cannot compete with carboxylate anions for the oxophilic Ln(III) cations [49]. This results in the coordination of six carboxylate ligands around each binuclear unit (to maintain the charge neutrality) and extension of the coordination network in three independent directions (3D) in the case of [Ln_2_(L)_2_(chdc)_3_]. When the nitrate anions are present in the reaction system, two carboxylate groups are substituted by the NO_3_^−^ anions, thus reducing the connectivity of the {Ln_2_(phen)_2_(NO_3_)_2_(OOCR)_4_} building unit and dimensionality of the coordination network to the layered 2D structure in **1**. The coordination net in the previously reported series [Ln_2_(phen)_2_(chdc)_3_]·½DMF is rather dense, with a theoretical pore volume of only 6% (*v/v*). On the contrary, the less entangled structures [Ln_2_(phen)_2_(NO_3_)_2_(chdc)_2_]·2DMF feature narrow channels ca. 3 × 7 Å^2^ running across the ABAB-packed coordination layers with calculated [50] guest accessible volume reaching 26% (*v/v*). The two series [Ln_2_(phen)_2_(chdc)_3_] and [Ln_2_(phen)_2_(NO_3_)_2_(chdc)_2_] represent a good example of the anion-controlled coordination topology and porosity in MOFs. The channels of the as-synthesized compounds [Ln_2_(phen)_2_(NO_3_)_2_(chdc)_2_] are occupied by the solvent DMF molecules, which could presumably be substituted to some other molecular substrates. 

### 2.2. Luminescent Properties 

Solid-state emission spectra (Figure 2a) were recorded for **1**, **2** and **3**. The spectrum of the Y(III)-based **1** possesses a broad emission band, typical to the complexes of d^0^ cations with phen-ligand-centered π*→π emission. The emission maximum appears at λ_max_ = 409 nm (λ_ex_ = 330 nm), indicating a blue color. The luminescence spectra of Eu(III)-based **2** and Tb(III)-based **3** compounds contain a series of narrow emission bands in the red and green areas, respectively, typical for such lanthanide cations. The spectrum of **2** (λ_ex_ = 340 nm) contains bands at 580 nm, 593 nm, 618 nm, 651 nm and 697 nm, which correspond to the series of ^5^D_0_ → ^7^F_J_ (J = 0, 1, 2, 3, 4) transitions in the Eu^3+^ cation. Both the ^5^D_0_ → ^7^F_2_:^5^D_0_ → ^7^F_1_ intensities ratio, which was calculated to be 4.68, and the presence of a ^5^D_0_ → ^7^F_0_ band indicated a low symmetry of the Eu^3+^ environment, fully consistent with the crystal structure. The luminescence spectrum of **3** (λ_ex_ = 330 nm) contains the characteristic emission bands at 490 nm, 545 nm, 586 nm and 622 nm, which correspond to the series of ^5^D_4_ → ^7^F_J_ (J = 6, 5, 4, 3) transitions in the Tb^3+^ cation.

The quantum yields (QY) for **1**–**3** at the abovementioned excitation wavelengths were determined as 2.4%, 32.0% and 13.5%, respectively, indicating limited to decent luminescence. Because the lanthanide cations feature rather low molar absorption, phen molecules apparently act as photosensitizers (“antennas”), which then transfer the absorbed energy to the emitting Ln^3+^ cations by a dipole–dipole energy transfer mechanism [51]. Such energy transfer is known to be quite efficient for Eu^3+^ and, to a lesser extent, for Tb^3+^ cations. On the contrary, the Y^3+^ cation has an electron-complete configuration with no f-orbitals; thus, such a ligand-to-metal energy transfer mechanism is not possible for **1**. Instead, the photoexcitation is released through a phen-centered π*→π transition, which is usually not very effective, as a great deal of the energy is released through a nonemitting relaxation along multiple vibrational states of the antenna ligand. Keeping in mind that the compounds **1**–**3** are completely isostructural, the coordination matrix **1** with low luminescence activity can be employed for the dilution of photoactive Eu^3+^ and/or Tb^3+^ cations in the homogeneous crystalline phase [52,53,54,55,56,57]. Such lanthanide doping of the yttrium framework could (i) improve the luminescence efficiency by reducing the concentration quenching known for such luminophores (Eu^3+^, Tb^3+^) and (ii) allow the mixing of the luminescence colors for more complex optical properties. Based on such an idea, a number of mixed-metal compositions containing small amounts of Tb^3+^ and/or Eu^3+^ cations in the Y(III)-based **1** were prepared and systematically investigated. 

As determined by ICP-MS, the actual lanthanide composition of all mixed-metal samples was close to the starting metal ratio, revealing no preferred inclusion of any Ln^3+^ cation (see Appendix A). The structural similarity and phase purity of the samples were confirmed by PXRD (Appendix A). The luminescence spectra of bimetallic compounds [Ln_2_(phen)_2_(NO_3_)_2_(chdc)_2_]∙2DMF (Ln_2_ = Y_1.998_Eu_0.002_ for **4**, Y_1.983_Eu_0.017_ for **5**, Y_1.998_Tb_0.002_ for **6**, Y_1.979_Tb_0.021_ for **7**) and trimetallic compounds (Ln_2_ = Y_1.813_Eu_0.006_Tb_0.181_ for **8**, Y_1.849_Eu_0.0014_Tb_0.1496_ for **9**, Y_1.855_Eu_0.049_Tb_0.096_ for **10**, Y_1.959_Eu_0.001_Tb_0.040_ for **11** and Y_1.9594_Eu_0.0002_Tb_0.0404_ for **12**) are shown in Appendix A, respectively. As could be expected, the resulting spectra for **4**–**12** are a superposition of the basic luminescence of the single-metal compounds **1**, **2** and **3**. 

The CIE 1931 diagrams assess the effective color of the luminescence and visualize the contribution of each color component. The corresponding CIE 1931 coordinates for the single-metals **1**, **2**, and **3**, as well as for the bimetal compounds **4**–**7**, are shown in Figure 2b and Appendix A. A strong contribution of the red component of Eu^3+^ is clearly observed, as the sample **4**, containing only 0.1 molar percent of europium(III) dopant in the yttrium(III) matrix, appears on the color diagram nearly in the middle between the pure Y^3+^ (**1**) and Eu^3+^ (**2**) samples, respectively. The contribution of the green component of Tb^3+^ is substantially lower, as the sample **7**, containing 1.0 molar percent of the terbium(III) dopant in the yttrium(III) matrix, still resides in the blue region. The luminescence quantum yields of the bimetallic samples **4**–**7** at λ_ex_ = 330 nm are in the range of 2.0–7.3% (Appendix A), manifesting a general trend for an increase in the QYs for the Eu^3+^/Tb^3+^-doped samples compared to that for the nondoped Y^3+^-based matrix **1** due to the contribution of the more efficient energy transfer mechanism discussed above. The observed relative degree of the color contribution Eu^3+^ > Tb^3+^ >> Y^3+^ is fully consistent with the QYs for the corresponding individual compounds. According to this dependence, several trimetallic samples were synthesized to further expand the variety of the emission colors, ultimately aiming to obtain a white luminescence with a balanced contribution of all three basic components. 

The CIE 1931 diagrams for various trimetallic compositions are shown in Figure 3 and Appendix A. The samples **8**–**10**, containing, in summary, 5–10% of the dopant metals in the Y^3+^ matrix, demonstrate a bright red or yellow emission with very high quantum yields from 30 to 84%, depending on the excitation wavelength λ_ex_ = 310–350 nm (Appendix A). The substantial increase in the luminescence efficiency of the Y(III)-diluted samples, compared with the pure Eu(III) or Tb(III) compounds, is explained by the concentration quenching, often observed when the fluorescent centers are situated within a few nanometers [58,59]. Obviously, the dilution by Y(III) spreads the average distance between the lanthanide cations within the coordination matrix and eliminates the concentration quenching. The obtained dopant concentration level seems to be optimal, as further reduction in the Eu^3+^/Tb^3+^ cations’ concentration to 2% in **11** and **12** significantly decreased the luminescence efficiency and the QYs down to a few percent. Additionally, the contribution of the blue component in the luminescence spectra of the trimetallic compounds **8**–**12** was found to depend strongly on the excitation energy. As clearly seen in Figure 3, the color coordinates on the CIE 1931 diagrams are consistently shifted from red or orange parts of the spectra towards the white color by increasing the excitation wavelength from λ_ex_ = 330 nm to λ_ex_ = 390 nm. For example, a practically daylight white color with a calculated color temperature (CT) value of 6819 K was achieved for **8** at λ_ex_ = 380 nm. A similar effect was observed earlier for the isostructural 3D MOFs [Ln_2_(bpy)_2_(chdc)_3_] and likely indicates a decrease in the efficiency of dipole–dipole ligand-to-lanthanide energy transfer at higher λ_ex_ [48]. Despite relatively low luminescence intensity of the near-white emission points, the obtained experimental data clearly demonstrate a successful color tuning in the studied system [Ln_2_(phen)_2_(NO_3_)_2_(chdc)_2_]·2DMF between blue, red and green emissions by a rational variation in the composition of the mixed-metal samples, as well as the excitation wavelength, showing a promising background for such materials in LED applications. 

### 2.3. Luminescent Sensing

The sufficient porosity of the coordination frameworks [Ln_2_(phen)_2_(NO_3_)_2_(chdc)_2_], coupled with their remarkable luminescence properties, suggest a further investigation of a guest-dependent luminescence for sensing applications. For example, the nitroaromatic organic molecules, being electron-deficient species, often quench the luminescence of the porous MOFs due to efficient charge transfer [60,61,62,63,64,65,66]. Preliminary experiments with single-metal compounds [Eu_2_(phen)_2_(NO_3_)_2_(chdc)_2_]∙2DMF (**2**) and [Tb_2_(phen)_2_(NO_3_)_2_(chdc)_2_]∙2DMF (**3**), wetted by the different liquids indicated modest dependence of the luminescence of **2** and several examples of noticeable quenching of the luminesce in case of **3**. For further characterization, the samples of **3** were immersed in liquid nitrobenzene, 2-nitrotoluene, 3-nitrotoluene or 4-nitro-*m*-xylene. These solvent-exchanged samples were characterized by IR spectroscopy (Appendix A) and powder X-ray diffraction (Appendix A), confirming the preservation of the overall framework structure. The solid-state luminescence measurements (Appendix A) for the guest-exchanged samples of **3** revealed that the quantum yields of the corresponding adducts are 3 to 12 times lower compared to the pristine **3** (Appendix A).

Very interesting results were obtained for the detection tests for several ketones and aldehydes, including cinnamal, in which efficient sensing has practical relevance. The liquid *o*-tolualdehyde, *p*-tolualdehyde, propional, 3-methybutanal, phenylacetaldehyde, acetophenone, 1,3-cyclohexenone and l(−)-carvone showed no visually observed quenching of the luminescence of **3**. On the contrary, an immersion of crystals **3** in liquid cinnamal resulted in fast, easily observable quenching. The adduct **3⊃cinnamal** was characterized by elemental CHN, thermogravimetric analysis, IR (Appendix A) and powder X-ray diffraction (Appendix A), confirming the guest exchange and preservation of the parent MOF structure. The luminescent measurements (Appendix A) confirmed the decrease of the QY down to 2.5%, indicating expressed quenching. During the next stage, diluted solutions of the cinnamal were probed in the detection experiments by suspensions of microcrystals of **3** in DMF. The microcrystalline **3** was prepared by continuous stirring of the reaction mixture to prevent a growth of larger crystallites. The microcrystalline suspension was stable for at least 20 min, allowing the systematic measurements of the luminescence properties. The corresponding excitation and emission spectra are shown in Figure 4a. First of all, the luminescence of the DMF suspension of the microcrystals **3** proved to be very efficient, as the apparent quantum yield reaches QY = 78%, much greater than that for the solid sample of the larger crystallites (13.5%). As the cinnamal concentration in the DMF is increased by two orders of magnitude from ~2.9 × 10^−3^ M (1:2700 dilution level or 0.037% *v/v*) to 2.4 × 10^−1^ M, the measured quantum yield is decreased from 36% to virtually 0% (see also Figure 4b). The twofold drop of the brightness of the emission of **3** (from QY = 78% to QY = 36%) for a very diluted cinnamal solution could be easily observed, thus, representing the first example of the simple, efficient and selective MOF-based luminescence sensor for this important chemical. 

A remarkable selective quenching of the luminescence of **3** by cinnamal among other carbonyls could be rationalized in terms of the molecular geometry. Indeed, the molecular structure of the *trans*-cinnamal molecule is flat, and its van der Waals dimensions (~3 × 7 Å^2^) fit well to the size of the slit-like channels in **3** (Figure 5). The molecular sizes of all other carbonyl molecules are larger in at least one dimension, including propional, which contains a tetrahedral sp^3^-carbon atom bound to three hydrogens. Such a sterical factor plausibly prevents the diffusion of the other species into the narrow channels of the crystal structure **3**. Elemental analysis data for the solid adduct **3⊃cinnamal** ([Tb_2_(phen)_2_(NO_3_)_2_(chdc)_2_]·1.1DMF·0.9C_9_H_8_O) unambiguously shows a considerable substitution of DMF by the cinnamal, fully confirming the hypothesis. Moreover, the sorption of the cinnamal by **3** is supported by the excitation spectra (Figure 4). The excitation spectrum of the suspension in pure DMF contains a wide several-moded band in the region 260–370 nm. Even at ≈10^−3^ M concentration of cinnamal in solution, the lower-wavelength mode of excitation becomes nearly completely quenched, leaving only the excitation in the region of 310–370 nm, which then gradually decreases at increasing concentrations of the substrate. Such behavior is reasonable, as free cinnamal in solutions has an intensive absorption band with the maximum at ca. 288 nm [41] and demonstrates a strong interaction between the adsorbed cinnamal guest molecules and the host luminescent framework. Note that clear DMF solutions of cinnamal with the same concentrations were used for a baseline correction.

## 3. Experimental Section

### 3.1. Reagents

Trans-1,4-cyclohexanedicarboxylic acid (H_2_chdc, >97.0%), 1,10-phenanthroline monohydrate (phen∙H_2_O, >98.0%), 2-nitrotoluene (>99.0%) and 3-nitrotoluene (>99.0%) were received from TCI (Tokyo, Japan). Y(NO_3_)_3_∙6H_2_O (99.9% REO) and Eu(NO_3_)_3_∙6H_2_O (99.9% REO) were received from Dalchem (Khabarovsk, Russia). *N*,*N*-dimethylformamide (DMF, reagent grade) and Tb(NO_3_)_3_∙5H_2_O (reagent grade) were received from Vekton (Saint Petersburg, Russia). *trans*-Cinnamaldehyde (99%) was received from Sigma Aldrich (St. Louis, MO, USA). 4-nitro-*m*-xylene (99%) was received from Acros Organics (Geel, Belgium). All reagents were used as received without further purification.

### 3.2. Instruments

IR spectra in KBr pellets were recorded in the range 4000−400 cm^−1^ on a Bruker Scimitar FTS 2000 spectrometer (Billerica, MA, USA). Elemental analysis was performed on a VarioMICROcube (Elementar Analysensysteme GmbH, Hanau, Germany) analyzer. Powder X-ray diffraction (PXRD) analysis was performed at room temperature on a Shimadzu XRD-7000 diffractometer (Cu-Kα radiation, λ = 1.54178 Å or Co-Kα radiation, λ = 1.78897 Å, Kyoto, Japan). Thermogravimetric analysis was carried out using a Netzsch TG 209 F1 Iris (Selb, Germany) instrument under Ar flow (30 cm^3^∙min^−1^) at a 10 K∙min^−1^ heating rate. Excitation and emission photoluminescence spectra were recorded with a Horiba Jobin Yvon Fluorolog 3 (Edison, NJ, USA) spectrofluorometer equipped with a 450 W power ozone-free Xe lamp, an R928/1860 PFR Technologies cooled photon detector with a PC177CE-010 refrigerated chamber and double grating monochromators. The spectra were corrected for source intensity and detector spectral response by standard correction curves. The absolute quantum yield was measured using a G8 (GMP SA, Baden, Switzerland) spectralon-coated integrating sphere, which was connected to a Fluorolog 3 spectrofluorimeter. Digital photographs were taken using a Hamamatsu C11924-211 (Hamamatsu, Japan) UV-LED module with λ_ex_ = 365 nm. ICP-MS analysis was carried out on an Agilent 8800 (Santa Clara, CA, USA) spectrometer. The samples were digested in a mixture of HCl 36% water solution and H_2_O_2_ 30% water solution, then diluted by water prior to ICP-MS analysis. Diffraction data for single crystals of **1** were obtained on an automated Agilent Xcalibur (Santa Clara, CA, USA) diffractometer equipped with an AtlasS2 area detector (graphite monochromator, λ(MoKα) = 0.71073 Å). Diffraction data for single crystals of **2** and **3** were collected on the “Belok” beamline [67,68] (λ = 0.7927 Å) of the National Research Center ‘Kurchatov Institute’ (Moscow, Russian Federation) using a Rayonix SX165 (Evanston, IL, USA) CCD detector. Details of single-crystal X-ray analysis and structure refinement are provided in Appendix A. CCDC 2098708–2098710 contain the supplementary crystallographic data for this paper. These data can be obtained free of charge from the Cambridge Crystallographic Data Center at https://www.ccdc.cam.ac.uk/structures/ (accessed on 25 August 2021).

### 3.3. Synthetic Procedures

Synthesis of [Y_2_(phen)_2_(NO_3_)_2_(chdc)_2_]·2DMF (**1**). A 76.6 mg (0.20 mmol) amount of Y(NO_3_)_3_∙6H_2_O, 39.6 mg (0.20 mmol) of phen·H_2_O and 68.8 mg (0.40 mmol) of H_2_chdc were mixed in a 10 mL glass vial and dissolved in 5.00 mL of DMF. The obtained solution was heated at 110 °C for 48 h. After cooling to room temperature, the obtained white precipitate was filtered off, washed with DMF and dried in air. Yield: 81.0 mg (71%). IR spectrum characteristic bands (KBr, cm^−1^): 3425 (w., br., νO–H); 3086 and 3065 (w., νCsp^2^–H); 2937 and 2858 (m., νCsp^3^–H); 1678 (m., νCO_amide_); 1598 and 1590 (s., νCOO_as_); 1425 (s., νCOO_s_); 1294 (s., νNO_nitrate_). Elemental analysis data, calculated for [Y_2_(phen)_2_(NO_3_)_2_(chdc)_2_]·2DMF (%): C, 48.1; H, 4.4; N, 9.8. Found (%): C, 48.1; H, 4.5; N, 9.7. TG data: 13% weight loss at 140 °C; calculated for 2DMF: 13%. Framework decomposition at 380 °C. Phase purity of the bulk compound was confirmed by powder X-ray diffraction (PXRD).

Synthesis of [Eu_2_(phen)_2_(NO_3_)_2_(chdc)_2_]·2DMF (**2**) was carried out similarly to the synthesis of **1** using 89.2 mg (0.20 mmol) of Eu(NO_3_)_3_∙6H_2_O. Yield: 102 mg (80%). IR spectrum characteristic bands (KBr, cm^−1^): 3429 (w., br., νO–H); 3077 and 3064 (w., νCsp^2^–H); 2935 and 2858 (m., νCsp^3^–H); 1681 (m., νCO_amide_); 1588 (s., νCOO_as_); 1428 (s., νCOO_s_); 1294 (s., νNO_nitrate_). Elemental analysis data, calculated for [Eu_2_(phen)_2_(NO_3_)_2_(chdc)_2_]·2DMF (%): C, 43.3; H, 4.0; N, 8.8. Found (%): C, 43.1; H, 3.7; N, 8.3. TG data: 10% weight loss at 130 °C; calculated for 2DMF: 11%. Framework decomposition at 380 °C. Phase purity of the bulk compound was confirmed by PXRD.

Synthesis of [Tb_2_(phen)_2_(NO_3_)_2_(chdc)_2_]·2DMF (**3**) was carried out similarly to **1** using 87.0 mg (0.20 mmol) of Tb(NO_3_)_3_∙5H_2_O. Yield: 104 mg (81%). IR spectrum characteristic bands (KBr, cm^−1^): 3433 (w., br., νO–H); 3082 and 3066 (w., νCsp^2^–H); 2935 and 2858 (m., νCsp^3^–H); 1680 (m., νCO_amide_); 1590 (s., νCOO_as_); 1428 (s., νCOO_s_); 1291 (s., νNO_nitrate_). Elemental analysis data, calculated for [Tb_2_(phen)_2_(NO_3_)_2_(chdc)_2_]·2DMF (%): C, 42.9; H, 3.9; N, 8.7. Found (%): C, 43.0; H, 4.0; N, 8.3. TG data: 11% weight loss at 120 °C; calculated for 2DMF: 11%. Framework decomposition at 380 °C. Phase purity of the bulk material was confirmed by PXRD. Powder samples of **3** were prepared by scaling up this synthetic method (5-fold multiplication) and carrying out the heating at continuous intensive stirring.

Synthesis of mixed-metal samples **4**–**12** was carried out similarly to the synthesis of **1** using the corresponding quantities of Y(III), Eu(III) and Tb(III) nitrates listed in Appendix A. The metal ratio in the solid products was determined by ICP-MS analysis, and the phase purity was confirmed by PXRD.

Synthesis of [Tb_2_(phen)_2_(NO_3_)_2_(chdc)_2_]·1.1DMF∙0.9C_9_H_8_O **(3⊃cinnamal)**. A 100 mg amount of **3** was immersed in 2.0 mL of liquid cinnamal, which was refreshed twice at a one-day interval. After **3** days, the solid was filtered and dried at reduced pressure. IR spectrum characteristic bands (KBr, cm^−1^): 3430 (w., br., νO–H); 3080 and 3066 (w., νCsp^2^–H); 2937 and 2857 (m., νCsp^3^–H); 1678 (m., νC=O); 1588 (s., νCOO_as_); 1429 (s., νCOO_s_); 1293 (s., νNO_nitrate_). Elemental analysis data, calculated for [Tb_2_(phen)_2_(NO_3_)_2_(chdc)_2_]·1.1DMF·0.9C_9_H_8_O (%): C, 46.0; H, 3.8; N, 7.4. Found (%): C, 45.9; H, 3.9; N, 7.4. TGA: weight loss steps at 140 °C (DMF) and 200 °C (cinnamal). Framework decomposition at 390 °C.

Preparation of suspensions. The powder samples of **3** (20.0 mg) were dispersed in 5.0 mL of DMF solutions of cinnamal with the concentrations listed in Figure 4 in glass flasks. No precipitation of **3** powder was observed on the bottom of the flask for at least 20 min after slight shaking. The obtained suspensions were transferred into cuvettes and analyzed by luminescent measurements. Clear DMF solutions of cinnamal with the same concentrations were used for a baseline correction. Digital photographs (Figure 4b) were taken using an 8-fold lower concentration of **3** powder (1.0 mg of **3** per 2.5 mL of cinnamal solution in DMF) and Hamamatsu C11924-211 UV-LED module with λ_ex_ = 365 nm.

## 4. Conclusions

To summarize, this work reports a synthesis, structural characterization and investigation of the functional properties of new porous metal–organic frameworks [Ln_2_(phen)_2_(NO_3_)_2_(chdc)_2_]·2DMF (Ln^3+^ = Y^3+^, Eu^3+^, Tb^3+^, as well as corresponding bi- or trimetallic mixtures; phen = 1,10-phenantroline; chdc^2−^ = *trans*-1,4-cyclohexanedicarboxylate), presenting the first example of MOFs based on well-known binuclear carboxylate complexes {Ln_2_L_2_(NO_3_)_2_(OOCR)_4_} (L = any N-donor chelate ligand). Rational variation of the metal composition affords a series of multicolored solid-state luminophores with high quantum yields up to 84%, including near-white emission applicable in light-emitting devices. The compound [Tb_2_(phen)_2_(NO_3_)_2_(chdc)_2_]·2DMF demonstrated a selective turn-off luminescence response to cinnamaldehyde (cinnamal) in diluted solutions with millimolar concentrations. The selectivity of luminescence quenching in the presence of the cinnamal vs. a broad range of other carbonyl compounds is attributed to the specific structure of the channels, the geometry of which fits to the size of the cinnamal molecule. The reported series of isostructural MOFs represents a multifunctional platform with remarkable luminescence properties and possible applications in display devices or the analytical detection of important chemicals.

## Figures and Tables

**Figure 1 molecules-26-05145-f001:**
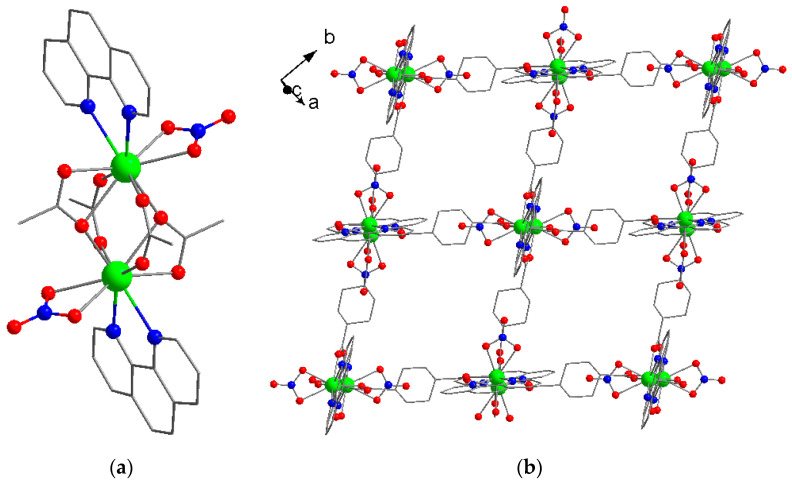
Binuclear block {Y_2_(phen)_2_(ONO_2_-κ^2^)_2_(μ-RCOO-κ^1^,κ^1^)_2_(μ-RCOO-κ^1^,κ^2^)_2_} in **1** (**a**). Coordination layer in **1** (**b**). Y atoms—green, O atoms—red, N atoms—blue. H atoms and quest molecules are not shown.

**Figure 2 molecules-26-05145-f002:**
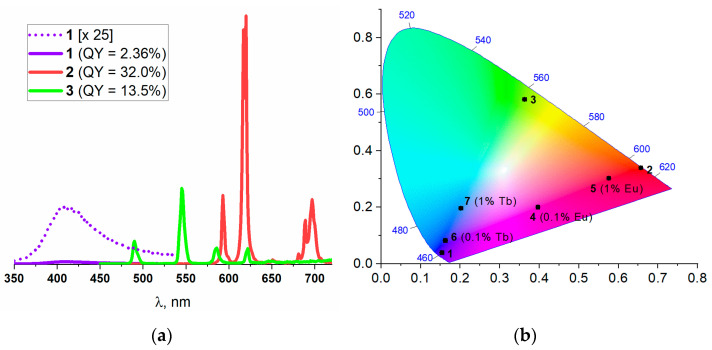
Emission spectra for individual compounds **1**–**3**, normalized on quantum yield (**a**). CIE 1931 chromaticity diagram for **1**–**3** and bimetallic compositions **4**–**7** at λ_ex_ = 330 nm (**b**).

**Figure 3 molecules-26-05145-f003:**
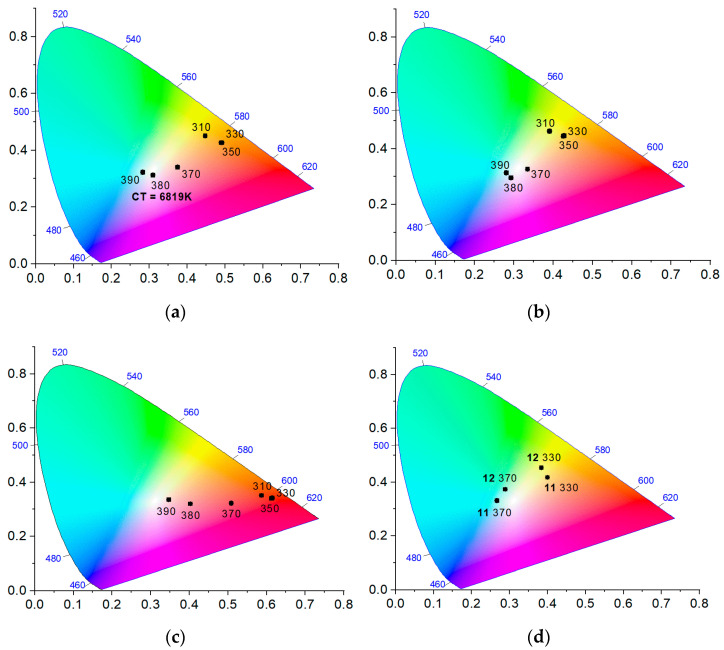
CIE 1931 chromaticity diagrams for trimetallic compositions **8** (**a**), **9** (**b**), **10** (**c**), **11** and **12** (**d**) at different excitation wavelengths.

**Figure 4 molecules-26-05145-f004:**
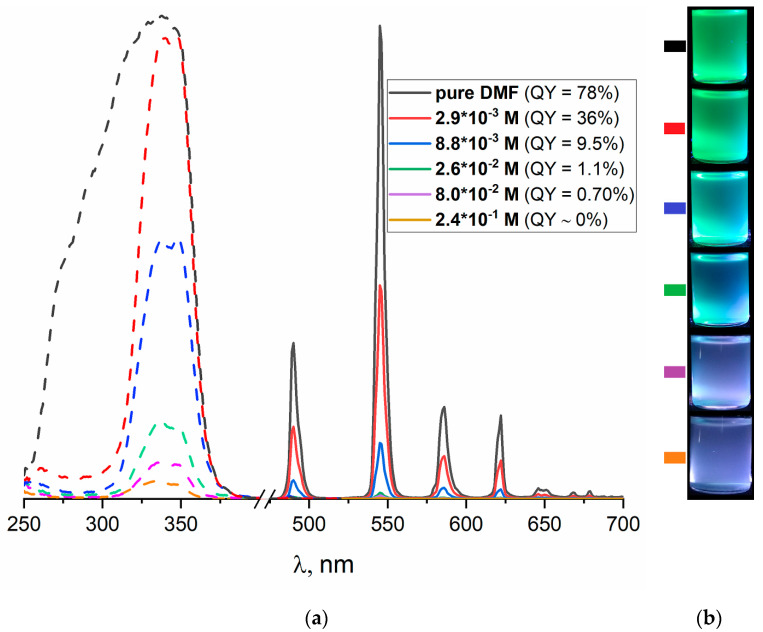
Excitation and emission (λ_ex_ = 340 nm) spectra for the suspensions of **3** in DMF solutions of cinnamal (**a**). Digital photographs of the suspensions of **3** with the corresponding cinnamal concentration under λ_ex_ = 365 nm (**b**).

**Figure 5 molecules-26-05145-f005:**
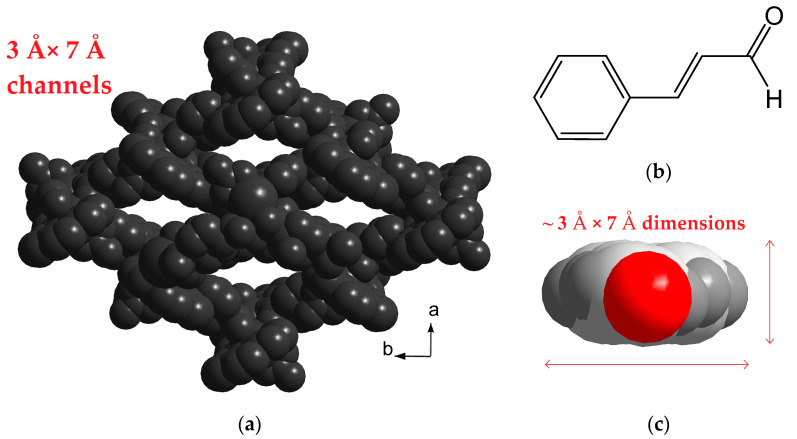
Three-dimensional (3D) packing of **3** in van der Waals spheres, viewed along the *c*-axis (**a**). Structural formula of cinnamal (**b**). Spatial representation of cinnamal molecule (**c**).

## Data Availability

CCDC 2098708–2098710 contain the supplementary crystallographic data for this paper. These data can be obtained free of charge from the Cambridge Crystallographic Data Cen-ter at https://www.ccdc.cam.ac.uk/structures/ (accessed on 25 August 2021).

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
