# Peer review of "Cinnamal Sensing and Luminescence Color Tuning in a Series of Rare-Earth Metal−Organic Frameworks with Trans-1,4-cyclohexanedicarboxylate"

_molecules, 2021, doi:10.3390/molecules26175145_

Round 1
Reviewer 1 Report
This work by Demakov and colleagues describes the preparation of isostructural lanthanides-based MOFs having interesting luminescent properties. Variations on metal composition afford multi-colored luminophores having huge quantum yields. These materials were employed for the cinnamal detection.
The work is well described and it has been rigorously carried out, but there are some minor points that the authors should add in any subsequent manuscript version:
- There are some references that the authors should add: (a) H. Furukawa, K. E. Cordova, M. O'Keeffe, O. M. Yaghi. Science 2013, 341, 1230444. https://doi.org/10.1126/science.1230444; (b) M. Ahmad, Y. Luo, C. Wöll, M. Tsotsalas, A. Schug. Molecules 2020, 25, 4875. https://doi.org/10.3390/molecules25214875; (c) V. F. Samanidou, E. A. Deliyanni. Molecules 2020, 25, 960. https://doi.org/10.3390/molecules25040960; (d) Y.-P. Wu, D.-S. Li, W. Xia, S.-S. Guo, W.-W. Dong. Molecules 2014, 19, 14352. https://doi.org/10.3390/molecules190914352; (e) E. Velasco, Y. Osumi, S. J. Teat, S. Jensen, K. Tan, T. Thonhauser, J. Li. Chemistry 2021, 3, 327. https://doi.org/10.3390/chemistry3010024; (f) O. M. Yaghi. ACS Cent. Sci. 2019, 5, 1295. https://doi.org/10.1021/acscentsci.9b00750
- Lines 57-80: the characteristics IR bands of the carboxylate-metal coordination should be commented.
- Lines 215-217: the characterization of the adduct should be compare to that of the pristine material. A TGA measurement of the adduct 3⊃cinnamal should be performed.
- CO2 adsorption experiments of the pristine MOF 3 and the adduct 3⊃cinnamal could be performed to observe changes in the porosity of the material. A decrease in the adsorbed gas volume is expected as a consequence of the formation of the cinnamaldehyde adduct.
Reviewer 2 Report
The manuscript describes the preparation and characterization of a range of MOFs from the lanthanides series namely yttrium, europium and terbium, and their bi- and trimetallic analogues. A complete characterization and resolving of the MOFs structure is provided. The as prepared monometallic MOFs showed color emissions in the RGB range which was tunable by just adjusting their doping level with combinations of the Y,Eu and Tb. Interestingly, the luminescence of the Tb-based MOF showed a selective significant quenching towards cinnamal in contrast to other aldehydes and ketones. The authors have done a nice piece of work and writing which worth to be published with only minor revision . Here are a few suggestions:
1) In the CIE 1931 diagrams, please add the x,y labels. Also, you could make a table with the coordinates (to be added in the supplementary Information) but I leave this to your preference/judgement.
2) An interesting work that could be mentioned in the introduction (maybe line 33-34) is the following DOI:10.1016/j.poly.2018.06.046 . This work highlights the tunability of the color emission of ion-exchangable Ca-MOFs with Eu and Tb ions towards white light emission. Again, I leave this to your judgement.
